# *Arabidopsis* V-ATPase d2 Subunit Plays a Role in Plant Responses to Oxidative Stress

**DOI:** 10.3390/genes11060701

**Published:** 2020-06-25

**Authors:** Shuang Feng, Yun Peng, Enhui Liu, Hongping Ma, Kun Qiao, Aimin Zhou, Shenkui Liu, Yuanyuan Bu

**Affiliations:** 1Key Laboratory of Saline-Alkali Vegetation Ecology Restoration, Northeast Forestry University, Ministry of Education, Harbin 150040, China; fengshuang86@163.com; 2College of Life Sciences, Northeast Forestry University, Harbin 150040, China; 3College of Horticulture and Landscape Architecture, Northeast Agricultural University, Harbin 150030, China; yunpeng0706@163.com (Y.P.); liuenhui1024@163.com (E.L.); hongpingma0818@163.com (H.M.); kunqiao@neau.edu.cn (K.Q.); aiminzhou@neau.edu.cn (A.Z.); 4The State Key Laboratory of Subtropical Silviculture, Zhejiang Agriculture and Forestry University, Lin’An 311300, Zhejiang, China; shenkuiliu@nefu.edu.cn

**Keywords:** V-ATPase d subunit, oxidative stress, sensitive, H^+^ flux, plasma membrane H^+^-ATPase, Arabidopsis

## Abstract

Vacuolar-type H^+^-ATPase (V-ATPase), a multisubunit proton pump located on the endomembrane, plays an important role in plant growth. The *Arabidopsis thaliana* V-ATPase d subunit (VHA-d) consists of two isoforms; AtVHA-d1 and AtVHA-d2. In this study, the function of *AtVHA-d2* was investigated. Histochemical analysis revealed that the expression of *AtVHA-d1* and *AtVHA-d2* was generally highly overlapping in multiple tissues at different developmental stages of Arabidopsis. Subcellular localization revealed that AtVHA-d2 was mainly localized to the vacuole. *AtVHA-d2* expression was significantly induced by oxidative stress. Analysis of phenotypic and H_2_O_2_ content showed that the *atvha-d2* mutant was sensitive to oxidative stress. The noninvasive microtest monitoring demonstrated that the net H^+^ influx in the *atvha-d2* roots was weaker than that in the wild-type under normal conditions. However, oxidative stress resulted in the H^+^ efflux in *atvha-d2* roots, which was significantly different from that in the wild-type. RNA-seq combined with qPCR analysis showed that the expression of several members of the plasma membrane H^+^-ATPase gene (*AtAHA*) family in *atvha-d2* was significantly different from that in the wild-type. Overall, our results indicate that AtVHA-d2 plays a role in Arabidopsis in response to oxidative stress by affecting H^+^ flux and *AtAHA* gene expression.

## 1. Introduction

The pH homeostasis in the endomembrane system is important for secondary active transport, cargo sorting, and protein trafficking [1,2,3,4]. Plants employ three classes of proton pumps to regulate cellular pH homeostasis: (i) plasma membrane H^+^-ATPase (P-ATPase); (ii) vacuolar H^+^- pyrophosphatase (V-PPase); and (iii) vacuolar-type H^+^-ATPase (V-ATPase) [1]. V-ATPase uses the energy released by ATP hydrolysis to transport protons (H^+^) across the membrane to regulate pH in the plant endomembrane system. V-ATPase is a highly conserved, multisubunit complex that consists of two domains, the integral membrane V_0_ domain, and the cytosolic V_1_ domain. The V_0_ domain, which consists of three to five different subunits of five subunits (VHA-a, VHA-c, VHA-c’’, VHA-d, VHA-e) on different organelles, is responsible for H^+^ translocation across membranes. The V_1_ domain, which consists of eight subunits (subunits A to H), is responsible for ATP hydrolysis [5,6]. The V_1_ domain is assembled into a complete enzyme by binding to the V_0_ domain; this binding is reversible in yeast [5]. The mechanism of this reversible dissociation in plants may be different from that in yeast [7].

Currently, it is considered that Arabidopsis V-ATPase is mainly located on *trans*-Golgi network/early endosome (TGN/EE) and vacuoles that are marked by the differential localization of three isoforms of AtVHA-a (AtVHA-a1, AtVHA-a2, and AtVHA-a3) [8,9]. The studies of *atvha-a1* single- and *atvha-a2/a3* double-mutant have shown that V-ATPase in the TGN/EE is required for exocytosis and recycling, whereas V-ATPase in the vacuole is required for effective nutrient storage [8,9,10]. In addition, V-ATPase plays a role in response to various stresses. For example, the growth of *deetiolated3* (*det3*), a mutant of *AtVHA-c*, is significantly inhibited and salt-sensitive [11,12]. The *atvha-a1* and *atvha-c5* mutants are also sensitive to salt [9,13]. Yeast two-hybrid analysis has shown that AtVHA-B interacts directly with salt overly sensitive 2 (AtSOS2) [12]. These studies suggest that V-ATPase may be involved in the plant salt stress response. Our previous studies have shown that expression of *AtVHA-c* genes is not only induced by salt stress but also by oxidative stress to varying degrees [14]. Other studies have shown that V-ATPase in yeast plays a role in the oxidative stress response [15,16]. In mammals, fungi, and plants, V-ATPases are subject to oxidative inactivation and activity can be recovered using reducing agents [17,18,19]. For example, *Arabidopsis* V-ATPase is reversibly inactivated by hydrogen peroxide (H_2_O_2_), and activity can be recovered using reducing agents [20]. However, the regulatory mechanism of V-ATPase in oxidative stress is not clear.

In the whole enzyme of V-ATPase, VHA-d is located on top of the ring formed by the six VHA-c subunits. Thus VHA-d participates in the control of either the reversible dissociation of V_0_ and V_1_ domains or coupling of proton transfer and ATP hydrolysis [5]. In yeast, VHA-d was found to be likely involved in the coupling of proton transport and ATP hydrolysis [21]. Furthermore, the structural stability of VHA-d depends on the formation of a disulfide bond [22]. In plants, only one study has shown that wheat (*Triticum aestivum* L.) *VHA-d* gene expression is induced by salt and abscisic acid [23]. The Arabidopsis *AtVHA-d* contains two isoforms, *AtVHA-d1* and *AtVHA-d2*, which are located in close proximity on chromosome three and may have arisen from a gene replication event [24]. At present, the expression pattern, subcellular localization, and function of the *AtVHA-d* genes are not clear. In this study, we investigated the expression patterns of *AtVHA-d* genes by promoter-driven *β*-glucuronidase (GUS) expression. The subcellular localization of AtVHA-d2 was observed by a green fluorescent protein (GFP) marker, the expression of *AtVHA-d* genes under various stresses was analyzed by quantitative real-time PCR (qPCR), and the phenotype and root H^+^ flux of the *atvha-d2* mutant under various stresses were investigated. Differentially expressed genes (DEGs) between the *atvha-d2* mutant and wild-type were identified and validated by RNA sequencing (RNA-seq) and qPCR.

## 2. Materials and Methods 

### 2.1. Plant Material and Growth Conditions

*Arabidopsis thaliana* ecotype Columbia-0 (Col-0) was used in this study, including wild-type, T-DNA insertion mutant, and transgenic plants. The *AtVHA-d2* T-DNA insertion mutant SAIL_141_G06 (CS806840) was obtained from the Arabidopsis Biological Resource Center (ABRC: http://www.arabidopsis.org/). Homozygous T-DNA insertion mutant plants were selected for PCR using the specific primers At3g28715-LP, At3g28715-RP, and left-border LB-3. The expression levels of *AtVHA-d2* and *AtVHA-d1* in mutant SAIL_141_G06 were analyzed by real-time qPCR using their respective primers (AtVHA-d1-qF, AtVHA-d1-qR, AtVHA-d2-qF, and AtVHA-d2-qR; Appendix A). The Arabidopsis seeds were surface sterilized and sown on 1/2 Murashige and Skoog (MS) medium supplemented with 1% agar and 3% sucrose (pH 5.8). The seeds were incubated at 4 °C for 3 d and then germinated in a growth chamber at 22 °C under a 12 h/12 h light/dark photoperiod (100 µmol m^−2^ s^−1^ light intensity).

### 2.2. Vector Construction and Arabidopsis Transformation

To construct *proAtVHA-d1*:*GUS* and *proAtVHA-d2*:*GUS*, 1644 and 1657 bp of the *AtVHA-d1* and *AtVHA-d2* promoter regions, respectively, were amplified using Col-0 genomic DNA and cloned into the HindIII/BamHI or KpnI/XhoI sites of pBI121-GUS or pBI121-GUS (modified, more enzyme digestion sites were added) vectors. To construct the *AtVHA-d2-GFP* fusion genes, the open reading frame of *AtVHA-d2*, without the stop codon, was amplified using PCR and cloned into the XbaI/KpnI sites of the pBI121-PutVHA-c-GFP vector, based on a previously reported method [14]. The constructs were stably transformed into *Arabidopsis* via the *Agrobacterium tumefaciens*-mediated floral dip method [25]. The T3 transgenic plants were identified by reverse transcription PCR. The specific primers used in this study are listed in Appendix A. 

### 2.3. Confocal Laser Scanning Microscopy

Roots of *Arabidopsis* seedlings stably expressing AtVHA-d2-GFP were washed twice with liquid 1/2 MS medium immediately before visualizing via confocal laser scanning microscopy (Nikon, A1, Tokyo, Japan). Using the 40× oil-immersion objective lens (Plan Apochromat; numerical aperture 1.3), the observation area was 1024 × 1024 pixels, and the pixel dwell time was 0.497 μs. GFP signals were detected using a 500–530 nm emission wavelength (FITC) after excitation with a 488 nm laser. 

### 2.4. Histochemical β-Glucuronidase Staining

Seedlings and different organs of transgenic Arabidopsis (*proAtVHA-d1:GUS* and *proAtVHA-d2: GUS*) were immersed in the staining buffer (100 mM sodium phosphate, pH 7.0, 10 mM EDTA, 0.5 mM K_3_[Fe(CN)_6_], 0.5 mM K_4_[Fe(CN)_6_], 0.1% Triton X-100) supplemented with 0.5 mM 5-bromo-4- chloro-3-indolyl-β-d-glucuronide (X-Gluc) for 12 h at 37 °C. Chlorophyll in green parts was removed by repeated washing in 95% ethanol.

### 2.5. qPCR Analyses

For abiotic stress treatments, 7-day-old *Arabidopsis* seedlings were exposed to 150 mM NaCl, 300 mM mannitol, 5 mM H_2_O_2_, 100 µM abscisic acid (ABA), cold (4 °C), or heat (37 °C). The seedlings were collected at different time points (0, 3, 6, 12, and 24 h) after treatment, and then frozen immediately in liquid nitrogen for RNA extraction. 

Total RNA was extracted using the RNAprep pure plant kit (Tiangen, Beijing, China), and cDNA was synthesized from 1 µg of total RNA using the M-MLV RTase cDNA synthesis kit (TaKaRa, Shiga, Japan), according to the manufacturer’s instructions. qPCR was performed on an Mx3000P QPCR system (Agilent Technologies, Palo Alto, CA, USA). The reaction components per 20 µL were as follows: 10 µL SYBR Green Mix (Agilent), 1 µL 10 µM of each primer and 1 µL cDNA, and 7 µL H_2_O. The thermal cycling program was as follows: initial denaturation at 95 °C for 120 s, and 40 cycles at 95 °C for 10 s, 60 °C for 30 s, and 72 °C for 30 s. The *Arabidopsis AtActin2* gene was used as an internal control [14]. The relative quantification of gene expression was evaluated using the delta-delta-Ct method. The transcript level in untreated seedlings (control) was set as 1.0. The primers used in this study are shown in Appendix A.

### 2.6. Stress Tolerant Phenotype 

For the stress tolerance assay, 30 seeds of Col-0 and the *atvha-d2* mutant were treated at 4 °C for 3 days and then grown vertically on 1/2 MS medium (control) or 1/2 MS medium supplemented with different concentrations of NaCl (75 and 100 mM), H_2_O_2_ (1 and 2 mM), or mannitol (175 and 200 mM). After 14 days, seedling phenotypes were photographed, and the root length, relative root length, and fresh weight of the seedlings were measured. The experiment was repeated three times.

### 2.7. Hydrogen Peroxide (H_2_O_2_) Content Measurement

Ten-day-old seedlings of Col-0 and the *atvha-d2* mutant were grown on 1/2 MS medium and transferred to 1/2 MS medium supplemented with H_2_O_2_ (1, 2, and 3 mM) for 48 h. The 0.1 g seedlings were used to measure H_2_O_2_ content. The H_2_O_2_ content was measured using the H_2_O_2_-1-Y assay kits (Comin, Suzhou, China), according to the manufacturer’s instructions.

### 2.8. Net H^+^ Flux Measurement

Net H^+^ flux was measured using the Noninvasive Microtest Technology (NMT100 Series, YoungerUSA LLC, Amherst, MA, USA) as described previously [13,26,27]. Seven-day-old seedlings of Col-0 and the *atvha-d2* mutant were grown on 1/2 MS medium, and were exposed to mannitol (200 mM), NaCl (100 mM), and H_2_O_2_ (2 mM) for 24 h. Root segments were immobilized in the measuring solution (0.1 mM KCl, 0.1 mM CaCl_2_, 0.1 mM MgCl_2_, 0.5 mM NaCl, and 0.3 mM MES, pH 5.8) to measure the H^+^ flux. The roots were fixed to the bottom of the plate using resin blocks and filter paper strips. Each sample was measured continuously for 10 min. To take flux measurements, the ion-selective electrodes were calibrated using pH 5.5, 6.0, and 6.5 solutions, respectively. The H^+^ flux rate, based on the voltages monitored between two points (0 and 20 µm), was calculated using iFluxes/imFluxes 1.0 software (Younger USA LLC, Amherst, MA, USA). The calibration slope for H^+^ was 54.42 mV/decade. Six biological repeats were performed for each analysis.

Ion flux was calculated by Fick’s law of diffusion:*J* = −*D* (*dc*/*dx*)
where *J* represents the ion flux in the x direction, *dc/dx* is the ion concentration gradient, and *D* is the ion diffusion constant in a particular medium.

### 2.9. RNA-seq and DEGs Analysis 

Seedlings of Col-0 and the *atvha-d2* mutant were treated in 1/2 MS medium supplemented with H_2_O_2_ (2 mM) for 0, 12, and 24 h. Total RNA from the seedlings was isolated using TRIzol reagent (Invitrogen, Carlsbad, CA, USA). Subsequently, the RNA samples were sent to the Beijing Genomic Institute (Shenzhen, China) for RNA-seq.

RNA-seq data processing and DEGs analysis as previously described [28,29]. DEGs were screened with a false discovery rate threshold of 0.01 and an absolute log2 ratio of 1. All DEGs were mapped to each term of the KEGG module from a Kyoto Encyclopedia of Genes and Genomes (KEGG) databases, and significant pathways were defined based on a corrected *p* ≤ 0.05.

### 2.10. Statistical Analysis

All experiments were conducted at least in three independent biological and three technical replicates. The data were analyzed using a one-way analysis of variance in SPSS (SPSS, Inc., Chicago, IL, USA), and statistically significant differences were calculated using the Student’s *t*-test, with *p* < 0.05 (*) and *p* < 0.01 (**) as the threshold for significance.

## 3. Results

### 3.1. Tissue Specificity of AtVHA-d Genes Expression

*AtVHA-d1* and *AtVHA-d2* were two highly similar genes, and they shared 99.4% identity in the amino acid sequence (Figure 1A). Transmembrane prediction using a hidden Markov model prediction showed that both AtVHA-d1 and AtVHA-d2 proteins had no transmembrane domain (Figure 1B). To verify whether functions of *AtVHA-d1* and *AtVHA-d2* are redundant, their expression patterns were investigated by promoter-driven GUS reporter transgene. During early seedling development, the expression of *proAtVHA-d1:GUS* and *proAtVHA-d2:GUS* was detected in all tissues of the seedlings, including roots, stems, leaves, and stipule primordia (Figure 2A,B). In mature plants, both *proAtVHA-d1:GUS* and *proAtVHA-d2:GUS* were expressed in flowers and siliques, including sepal, anther, and embryo sac (Figure 2C,D). In conclusion, *AtVHA-d1* and *AtVHA-d2* genes are broadly expressed in plant tissues and exhibit overlapping expression patterns.

### 3.2. Subcellular Localization of AtVHA-d2 

Only two amino acids of AtVHA-d1 and AtVHA-d2 were different; thus the subcellular localization of one member (AtVHA-d2) in Arabidopsis was investigated using GFP as a fusion protein marker. The confocal images showed that the GFP signals were mainly localized to the vacuoles in root cells of wild-type Arabidopsis seedlings stably expressing AtVHA-d2-GFP (Figure 3).

### 3.3. Sensitivity of the Atvha-d2 Mutant to Multiple Stresses 

qPCR analysis showed that the expression of *AtVHA-d1* was not affected by salt, osmotic, oxidative, and ABA stress, but *AtVHA-d2* was significantly induced by these stressors. Under cold and heat treatment, the expression of both *AtVHA-d1* and *AtVHA-d2* was not obviously affected (Figure 4A,B). The result suggests that *AtVHA-d2* may play a role in the *Arabidopsis* response to multiple stresses. 

To better understand the role of *AtVHA-d2* in response to multiple stresses, the T-DNA insertion mutant of *AtVHA-d2* was identified. The genomic PCR and sequencing analysis showed that T-DNA was inserted into the fifth exon of the *AtVHA-d2* gene (Figure 5A–C; Appendix A). qPCR analysis revealed that the *AtVHA-d2* mRNA level in *atvha-d2* was approximately 10% of that in Col-0. However, the *AtVHA-d1* mRNA level in *atvha-d2* was similar to that in Col-0 (Figure 5D). 

The phenotypes of the *atvha-d2* mutant and Col-0 were compared under normal and multiple stress conditions. On 1/2 MS medium or 1/2 MS medium supplemented with NaCl (75 and 100 mM), H_2_O_2_ (1 and 2 mM), and mannitol (175 and 200 mM), the primary root length of the *atvha-d2* mutant was generally lower than that of Col-0 (Figure 6A,B). Under H_2_O_2_ and mannitol stress, the fresh weight of the *atvha-d2* seedlings was significantly lower than that of Col-0, and there were no significant differences in the normal conditions (1/2 MS medium; Figure 6C). The relative root length analysis showed that the inhibition ratio of H_2_O_2_ on the primary root growth of the *atvha-d2* mutant was significantly higher than that of Col-0 (Figure 6D). Moreover, the H_2_O_2_ accumulation level in the *atvha-d2* mutant was significantly higher than that in Col-0 under both normal and H_2_O_2_ stress (Figure 7). These results suggest that the *atvha-d2* mutant is sensitive to oxidative stress.

### 3.4. H^+^ Flux in Root of the Atvha-d2 Mutant under Multiple Stresses 

The net H^+^ flux in the roots of the *atvha-d2* and Col-0 seedlings was monitored using NMT. The H^+^ influx in the roots of *atvha-d2* and Col-0 seedlings grown on 1/2 MS medium was observed, but the rate of H^+^ influx in *atvha-d2* was significantly lower than that in Col-0 (Figure 8A,B). The result indicated that H^+^ flux in the *atvha-d2* roots is impaired. Furthermore, the H^+^ flux in the roots of *atvha-d2* and Col-0 was compared under multiple stresses. Similar to that in untreated roots (1/2 MS medium), the mannitol treatment had no obvious effect on H^+^ flux in the *atvha-d2* and Col-0 roots. The NaCl treatment resulted in H^+^ efflux in both *atvha-d2* and Col-0 roots, but there was no significant difference between them. Under the H_2_O_2_ treatment, a H^+^ influx in the Col-0 roots was observed; however, in the *atvha-d2* roots, a H^+^ efflux was observed (Figure 8A,B). The result indicates that H_2_O_2_ treatment significantly affects H^+^ flux in the *atvha-d2* roots.

### 3.5. Identification of DEGs between Atvha-d2 and Wild-Type under Normal and Oxidative Stress Conditions

To identify DEGs between the *atvha-d2* mutant and Col-0, RNA-seq was performed. Under normal and oxidative stress conditions, 278 DEGs (Appendix A) were identified (Figure 9A,B). These DEGs were most significantly enriched in the V-type ATPase, eukaryotes module from the KEGG database (Figure 9C). This module contains 28 genes (Appendix A) that, with the exception of *AtVHA-d2*, were upregulated in the *atvha-d2* mutant (Figure 9D). The result showed that the inhibition of *AtVHA-d2* expression results in an upregulation of the expression of other V-ATPase assembly subunits under normal and oxidative stress conditions. Moreover, the expression of genes encoding P-ATPase (*AtAHA*) and V-PPase (*AtAVP*) in Col-0 and *atvha-d2* were different under normal and oxidative stress conditions (Figure 9E,F).

### 3.6. Expression of AtAHA and AtAVP Genes in atvha-d2 and Wild-Type under Normal and Oxidative Stress Conditions

The expression of 11 *AtAHA* and two *AtAVP* genes in the *atvha-d2* mutant and Col-0 under normal and oxidative stress conditions was compared by qPCR. Among the 11 *AtAHA* genes, the expression of *AtAHA1*, *AtAHA2*, and *AtAHA7* in the *atvha-d2* mutant was significantly lower than that in Col-0, whereas that of *AtAHA4*, *AtAHA5*, and *AtAHA8* was significantly higher than that in Col-0 under normal conditions. After 24 h of oxidative stress, the expression of *AtAHA1*, *AtAHA2*, *AtAHA4*, and *AtAHA5* in the *atvha-d2* mutant was significantly higher than that in Col-0 (Figure 10A). Two *AtAVP* genes, *AtAVP1* and *AtAVP2*, did not show significant changes in expression between Col-0 and the *atvha-d2* mutant (Figure 10B). The results showed that the inhibition of the *AtVHA-d2* expression causes changes in the expression of most members of *AtAHA* genes under normal and oxidative stress conditions.

## 4. Discussion

*AtVHA-d1* and *AtVHA-d2* are highly similar in sequence (Figure 1A), and it is presumed that they may have arisen from a gene replication event [24]. Moreover, their expression pattern is highly overlapping. The promoter-driven GUS expression analysis showed that *AtVHA-d1* and *AtVHA-d2* were generally expressed in multiple tissues at different developmental stages of *Arabidopsis*, including all tissues of seedlings, especially the stipule primordia, as well as pollens and embryo sacs (Figure 2), indicating that they are nonredundant. In addition, the tissue specificity of the expression of *AtVHA-d* genes is highly consistent with that of *AtVHA-c* genes reported previously [14,30]. VHA-d and VHA-c are essential subunits of the complete V-ATPase enzyme [6]. VHA-d itself does not have a transmembrane domain; it can be located on the membrane by combining with VHA-c [5]. As was expected, confocal observation showed that AtVHA-d2 was mainly localized to the vacuoles (Figure 3). Furthermore, the localization pattern of AtVHA-d2 was similar to that of AtVHA-c5 [13]. The consistency of the expression pattern and subcellular localization suggests that AtVHA-d2, together with AtVHA-c, may be involved in the assembly of the V-ATPase in the expressed tissues. However, our observations cannot completely exclude AtVHA-d2-GFP localization on other endomembranes.

Studies have shown that V-ATPase is involved in multiple stress responses [12,23,31]. The expression of *AtVHA-d* genes was investigated under abiotic stress conditions that significantly inhibited the growth of Arabidopsis seedlings, including 150 mM NaCl, 300 mM mannitol, 5 mM H_2_O_2_, 100 µM ABA, cold (4 °C), and heat (37 °C). qPCR analysis showed that *AtVHA-d2* was significantly induced by multiple stresses, especially oxidative stress (Figure 4). The phenotype of the *atvha-d2* mutant and wild-type were compared under the conditions of abiotic stress that moderately inhibited the growth of Arabidopsis seedlings, including 75 and 100 mM NaCl, 175 and 200 mM mannitol, and 1 and 2 mM H_2_O_2_. Phenotypic analysis showed that the relative root length and fresh weight of *atvha-d2* were significantly lower than those of Col-0 under oxidative stress (Figure 6). Furthermore, the H_2_O_2_ content in *atvha-d2* was higher than that in Col-0 under normal conditions and oxidative stress (Figure 7). Under normal culture conditions (1/2 MS), H^+^ influx occurred in the *atvha-d2* roots, whereas H_2_O_2_ treatment caused H^+^ efflux, which was significantly different from that in Col-0 (Figure 8). These results suggest that *atvha-d2* is sensitive to oxidative stress, which may be associated with abnormal H^+^ flux in roots. The extrusion of H^+^ in roots is directly regulated by P-ATPase [32]. In Arabidopsis, P-ATPase is encoded by 11 genes, *AtAHA1* to *AtAHA11*, of which *AtAHA1* and *AtAHA2* are the most highly expressed isoforms [32,33]. RNA-seq combined with qPCR analysis showed that the expression of *AtAHA1* and *AtAHA2* in *atvha-d2* was significantly higher than that in Col-0 after oxidative stress (Figure 9E and Figure 10A). The results suggest that the H^+^ efflux in *atvha-d2* roots under oxidative stress (for 24 h) may be caused by the higher expression of *AtAHA1* or *AtAHA2*. V-ATPase together with P-ATPase and V-PPase, co-regulates the cytosolic pH in the plant cells [1,3]. Compared to that in Col-0, the expression of genes encoding the other V-ATPase assembly subunits, P-ATPase and V-PPase in the *atvha-d2* mutant were altered, as shown RNA-seq and qPCR (Figure 9 and Figure 10). The result suggests that V-ATPase, P-ATPase, and V-PPase play synergistic roles in H^+^ balance through gene expression regulation. For example, the expression of *AtVHA-c*, *AtAVP1*, *AtAHA1*, and *AtAHA2* is simultaneously affected in the *atsos1* mutant (SOS1, a plasma membrane Na^+^/H^+^-antiporter) [34].

## 5. Conclusions

In conclusion, our results indicate that AtVHA-d2 plays a role in Arabidopsis in response to oxidative stress by affecting H^+^ flux, which may be related to differential expression of *AtAHA* genes.

## Figures and Tables

**Figure 1 genes-11-00701-f001:**
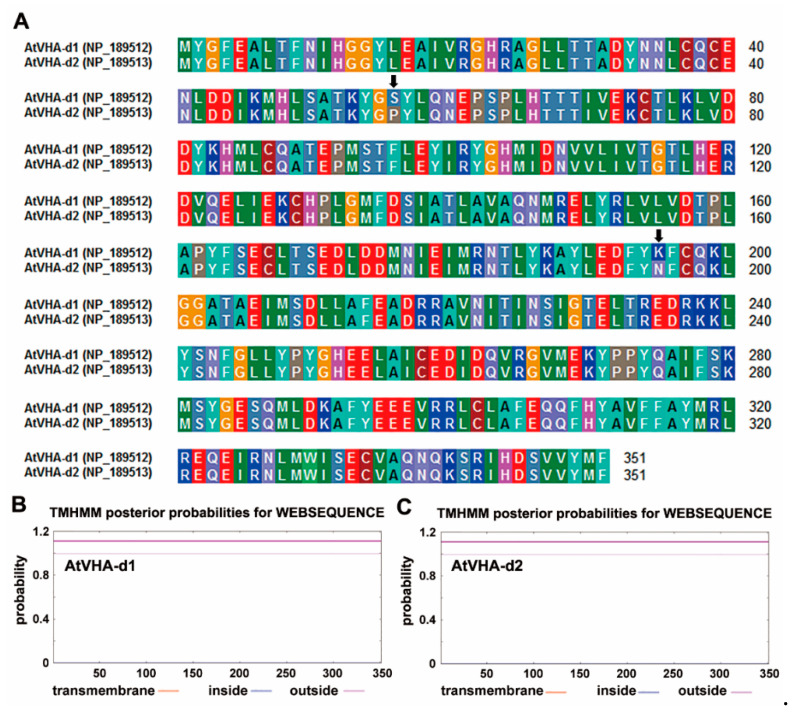
Sequence analysis of AtVHA-d1 and AtVHA-d2 proteins. (**A**) Amino acid sequence alignment of AtVHA-d1 and AtVHA-d2. The same color residues indicate identical residues in each sequence. The arrows indicate two different amino acids. Transmembrane domain prediction of AtVHA-d1 (**B**) and AtVHA-d2 (**C**).

**Figure 2 genes-11-00701-f002:**
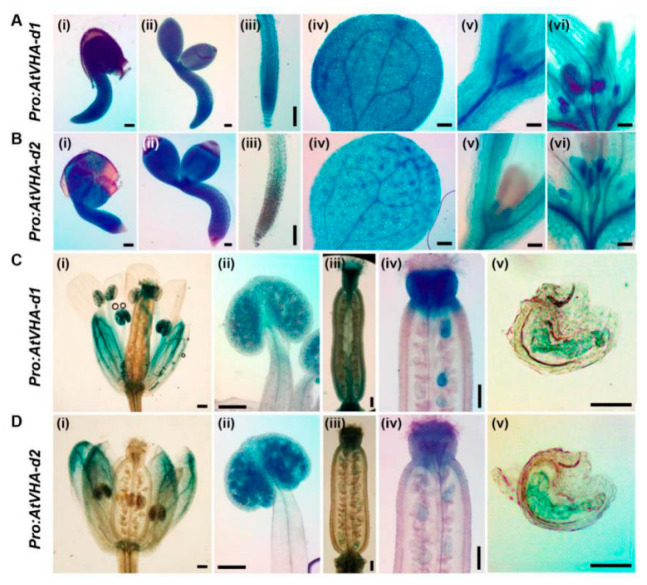
*AtVHA-d1* and *AtVHA-d2* gene promoter-GUS expression in Arabidopsis. Histochemical GUS staining was carried out at different development stages and in various tissues of *AtVHA-d1* and *AtVHA-d2* genes promoter-GUS transgenic Arabidopsis plants. *ProAtVHA-d1:GUS* (**A**) and *ProAtVHA-d2:GUS* (**B**) expression in all tissues of the seedlings (i and ii), root (iii), leaf (iv), and stipule primordia (v and vi); *ProAtVHA-d1:GUS* (**C**) and *ProAtVHA-d2:GUS* (**D**) expression in the sepal (i), anther (ii), base and top of the silique (iii and iv), and embryo sac (v). Scale bar = 200 µm.

**Figure 3 genes-11-00701-f003:**
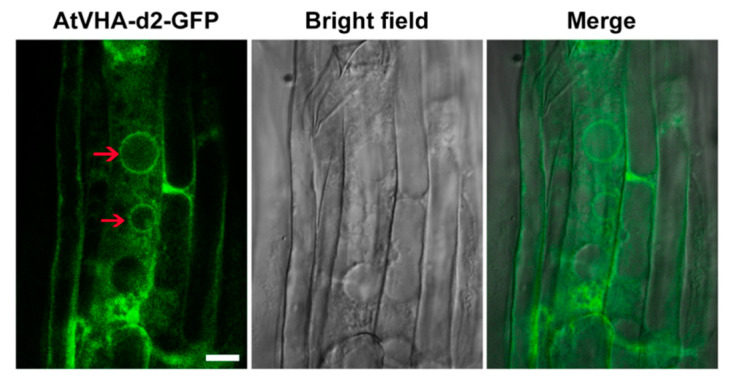
Subcellular localization of AtVHA-d2-GFP in *Arabidopsis*. Green fluorescent protein (GFP) fluorescence is green. Merge was created by merging the GFP and bright-field images. The red arrows indicate small vacuoles. Scale bar = 10 µm.

**Figure 4 genes-11-00701-f004:**
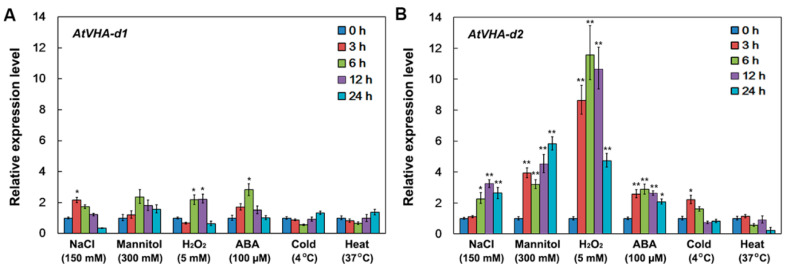
Expression analysis of *AtVHA-d1* and *AtVHA-d2* under multiple stresses. Ten-day-old *Arabidopsis* seedlings were treated with 150 mM NaCl, 300 mM mannitol, 5 mM H_2_O_2_, 100 µM ABA, 4 °C (cold), or 37 °C (heat) for 0, 3, 6, 12, and 24 h. The expression of *AtVHA-d1* (**A**) and *AtVHA-d2* (**B**) was investigated by qPCR. The *AtActin2* gene was used as an internal control, and the transcript level in untreated seedlings was set as 1.0. Asterisks indicate a significant difference between untreated and stress-treated seedlings (* *p* < 0.05; ** *p* < 0.01; Student′s *t*-test). Error bars represent the *SD* (*n* = 3).

**Figure 5 genes-11-00701-f005:**
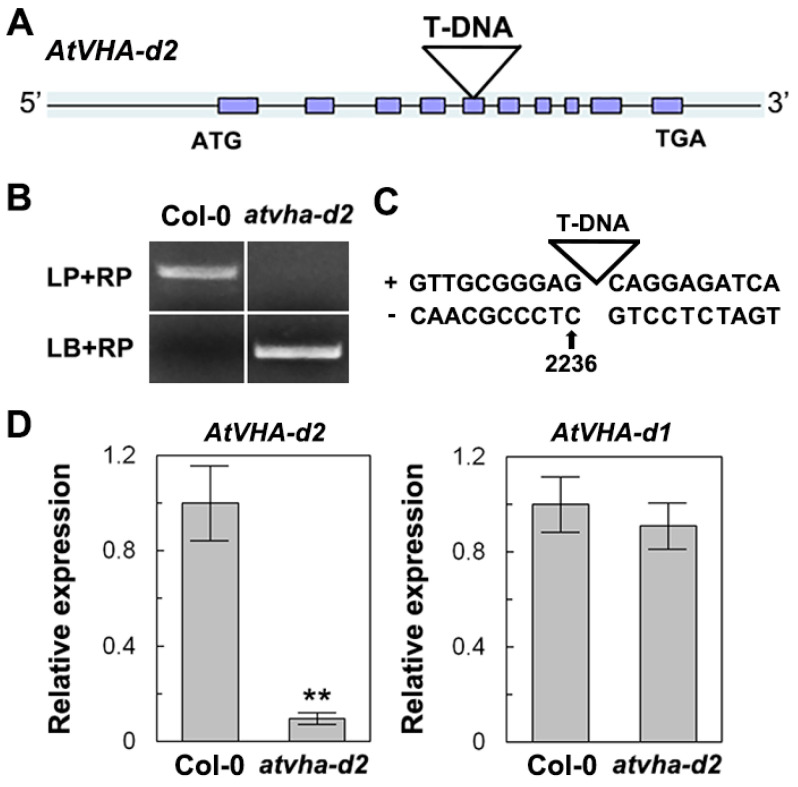
Identification of the Arabidopsis *atvha-d2* mutant. Schematic representation of *AtVHA-d2* (**A**) and genotyping (genomic DNA) (**B**). Original PCR detection pictures are shown in Appendix A. (**C**) The position of the T-DNA insertion in the *atvha-d2* mutant. **(****D)** Relative expression (mRNA) analysis of *AtVHA-d2* and *AtVHA-d1* in the *atvha-d2* mutant by qPCR. Asterisks indicate a significant difference between Col-0 and *atvha-d2* plants (** *p* < 0.01; Student’s *t*-test). Error bars represent the *SD* (*n* = 3).

**Figure 6 genes-11-00701-f006:**
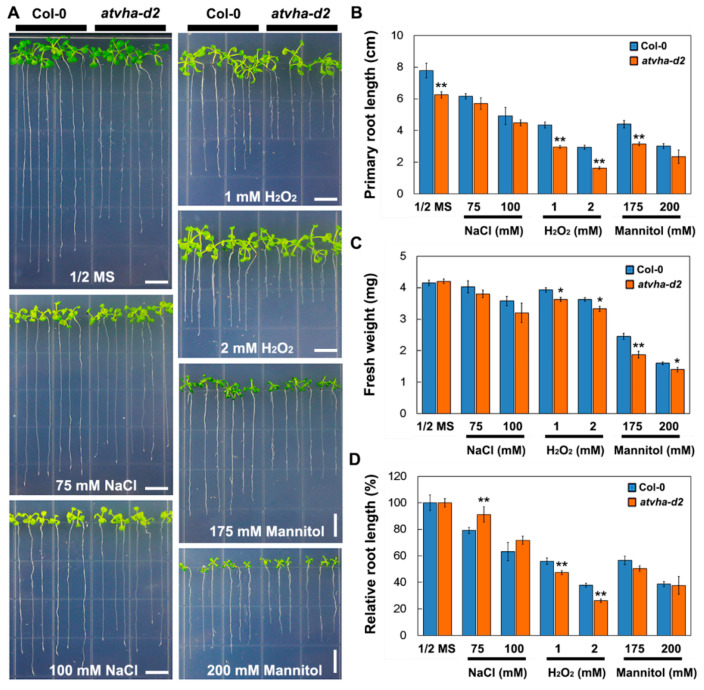
Comparison of phenotypes of Col-0 and the *atvha-d2* mutant under multiple stresses. Phenotypes (**A**), primary root length (**B**), relative root length (**C**), and fresh weight (**D**) of Col-0 and the *atvha-d2* mutant grown on vertical plates containing 1/2 MS or 1/2 MS medium with NaCl (75 and 100 mM), H_2_O_2_ (1 and 2 mM), and mannitol (175 and 200 mM) for 14 d. Asterisks indicate a significant difference between Col-0 and *atvha-d2* plants (* *p* < 0.05; ** *p* < 0.01; Student’s *t*-test). Error bars represent the *SE* (*n* = 3). Scale bar = 1 cm.

**Figure 7 genes-11-00701-f007:**
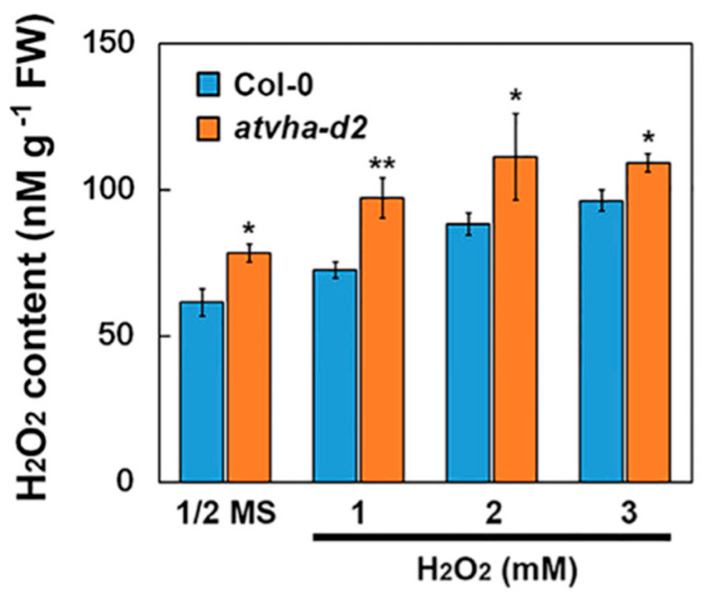
H_2_O_2_ content in Col-0 and *atvha-d2* mutant seedlings treated in 1/2 MS and 1/2 MS medium with H_2_O_2_ (1, 2, and 3 mM) for 48 h. Asterisks indicate a significant difference between Clo-0 and *atvha-d2* plants (* *p* < 0.05; ** *p* < 0.01; Student’s *t*-test). Error bars represent the *SE* (*n* = 3).

**Figure 8 genes-11-00701-f008:**
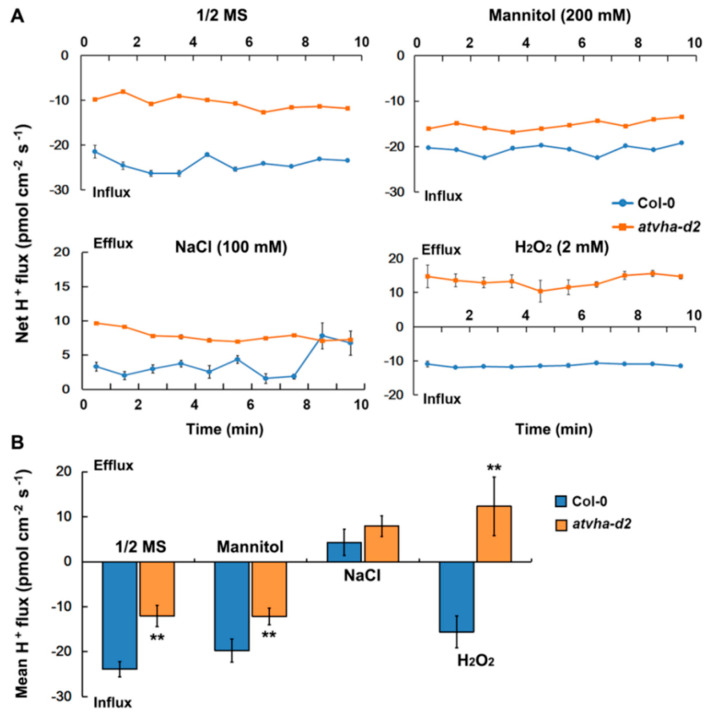
Net H^+^ flux in the roots of Col-0 and the *atvha-d2* mutant under multiple stresses. Net (**A**) and mean (**B**) H^+^ flux in the root elongation zone of Col-0 and *atvha-d2* seedlings (7-days-old) treated in 1/2 MS and 1/2 MS medium with mannitol (200 mM), NaCl (100 mM), and H_2_O_2_ (2 mM) for 24 h. Continuous flux was recorded for 10 min. Mean H^+^ flux from six samples (*n* = 6). Asterisks indicate a significant difference between Col-0 and *atvha-d2* plants (** *p* < 0.01; Student’s *t*-test). Error bars represent the *SE* (*n* = 6).

**Figure 9 genes-11-00701-f009:**
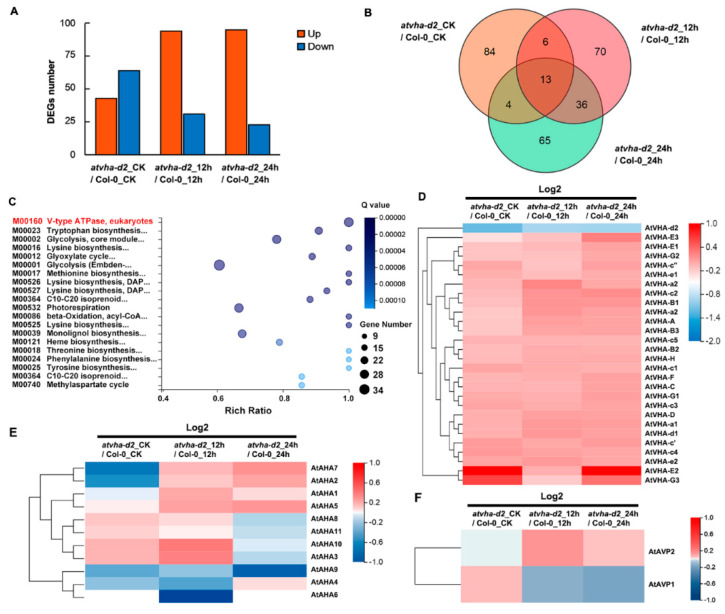
RNA-seq analysis of differentially expressed genes (DEGs) between Col-0 and the *atvha-d2* mutant under normal and oxidative stress. The number (**A**) and Venn diagram (**B**) of DEGs between Col-0 and the *atvha-d2* mutant under normal and oxidative stress conditions. DEG screening thresholds using false discovery rate ≤ 0.01 and the absolute value of log2Ratio ≥ 1. (**C**) KEGG module enrichment analysis of DEGs and their interaction genes. The rich factor is the ratio of the number of DEGs annotated in a given module term to the number of all genes annotated in the module term. The *q*-value is the corrected *p*-value and ranges from 0 to 1, and a lower *q*-value indicates greater intensity. Expression of V-ATPase subunits (*AtVHA*) (**D**), P-ATPase (*AtAHA*) (**E**), and V-PPase (*AtAVP*) (**F**) genes in Col-0 and the *atvha-d2* mutant under normal and oxidative stress conditions. Red rectangles represent the upregulation of genes, whereas green rectangles represent down-regulation. CK: control untreated samples.

**Figure 10 genes-11-00701-f010:**
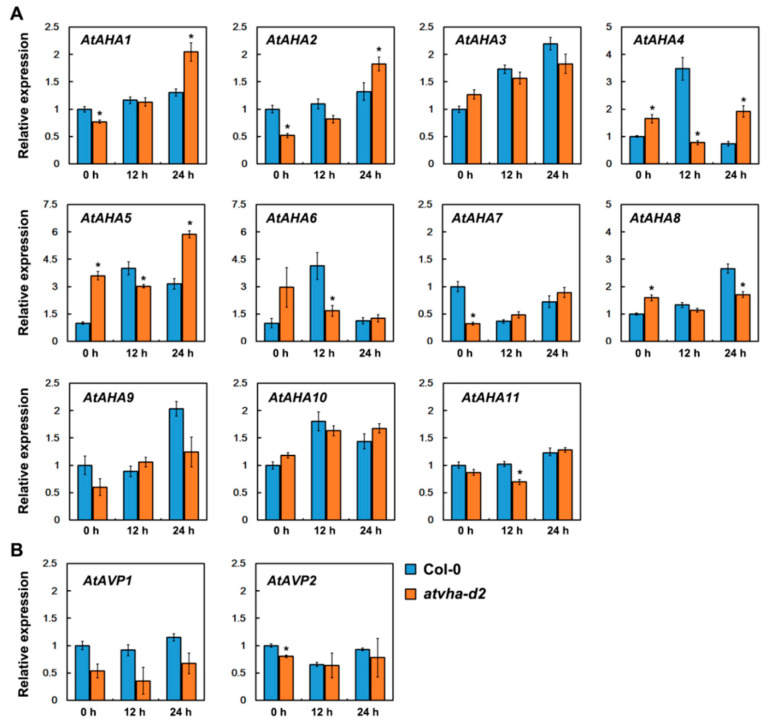
qPCR analysis of P-ATPase (*AtAHA*) and V-PPase (*AtAVP*) gene expression in Col-0 and the *atvha-d2* mutant under normal and oxidative stress conditions. The P-ATPase gene family contains 11 members (*AtAHA1* to *AtAHA11*) (**A**) and the V-PPase contains two members (*AtAVP1* and *AtAVP2*) (**B**). Asterisks indicate a significant difference between Col-0 and the *atvha-d2* plants (* *p* < 0.05; Student’s *t*-test). Error bars indicate the *SD* (*n* = 3).

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
