# Peer review of "Arabidopsis V-ATPase d2 Subunit Plays a Role in Plant Responses to Oxidative Stress"

_genes, 2020, doi:10.3390/genes11060701_

Round 1
Reviewer 1 Report
How many seeds were sown? How many replications? Fig. 1 needs better resolution, lines in B and C are not clear. Statistical analysis (Student test) are not explained in Methods. Scale bars missing in Fig. 2. Fig. 4 missing significant difference between variants. Fig.6 - A missing scale bars, BCD asterix are not readible. Fig. 9 - C not readible. Conclusions are very poor.
Author Response
15/06/2020
Genes
Dear Editor:
We wish to resubmit our manuscript entitled “Arabidopsis V-ATPase d2 Subunit Plays a Role in Plant Responses to Oxidative Stress” (manuscript ID: genes-830882). We are indeed thankful for the valuable comments, which were very helpful in improving our manuscript. We have perused the comments carefully and have made the suggested corrections.
We are hopeful that the revised manuscript meets your approval. The revised portions are marked in red. The major corrections done in the manuscript and the point-wise response to the reviewers’ comments are presented at the end of this letter.
We have made an earnest effort to incorporate all the suggested changes in the revised manuscript and look forward to a favorable decision from you.
Sincerely,
[Author’s name] Shuang Feng and Yuanyuan Bu
[Affiliation] Key Laboratory of Saline-Alkali Vegetation Ecology Restoration (Northeast Forestry University), Ministry of Education, Harbin 150040, PR China
[Postal address] Hexing Road No. 26, Xiangfang District, Harbin City, Heilongjiang Province 150040, China.
[Phone number] +86 451 8219 2763
[Fax number] +86 451 8219 2763
[Email address] fengshuang86@163.com; yuanyuanbu@nefu.edu.cn
Response to the reviewers’ comments
Open Review #1
Comments and Suggestions for Authors
- Comment:How many seeds were sown? How many replications? Fig. 1 needs better resolution, lines in B and C are not clear. Statistical analysis (Student test) are not explained in Methods. Scale bars missing in Fig. 2. Fig. 4 missing significant difference between variants. Fig.6-A missing scale bars, BCD asterix are not readible. Fig. 9 - C not readible. Conclusions are very poor.
Response: Thank you for your comment.
1) Thirty seeds were sown for each analysis. The experiment was repeated three times. We have added this information to Materials and Methods. (Page 3, lines 125-130).
“For the stress tolerance assay, 30 seeds of Col-0 and the atvha-d2 mutant were treated at 4 °C for 3 days and then grown vertically on 1/2 MS medium (control) or 1/2 MS medium supplemented with different concentrations of NaCl (75 and 100 mM), H2O2 (1 and 2 mM), or mannitol (175 and 200 mM). After 14 days, seedling phenotypes were photographed, and the root length, relative root length, and fresh weight of the seedlings were measured. The experiment was repeated three times.”
2) We have revised Figure 1 B and C, and its legend to improve its readability. (please see the new Figure 1; Page 5, lines 172-175).
3) We have added a description of "Statistical analysis" to the Materials and Methods. (Page 4, lines 155-159).
“2.10. Statistical analysis
All experiments were conducted at least in three independent biological and three technical replicates. The data were analyzed using a one-way analysis of variance in SPSS (SPSS, Inc., Chicago, IL, USA), and statistically significant differences were calculated using the Student’s t-test, with P < 0.05 (*) and P < 0.01 (**) as the threshold for significance.”
4) We have added scale bars to Figure 2. (please see the new Figure 2; Page 6, line 176-181).
5) We have added an asterisk representing the significance analysis in Figure 4. (please see the new Figure 4; Page 7, lines 197-203).
6) We have added scale bars to Figure 6A, and magnified the asterisks in Figure 6B,C,D. (please see the new Figure 2; Page 8, lines 226-231).
7) We have revised Figure 9C, to improve its readability. (please see the new Figure 2; Page 10, lines 262-272).
8) We have revised the conclusion. (Pages 12, 13, lines 332-342).
“5. Conclusions
In this study, the function of Arabidopsis AtVHA-d genes was characterized. Histochemical analysis revealed the expression of AtVHA-d1 and AtVHA-d2 was generally highly overlapping in multiple tissues of Arabidopsis. qPCR analysis showed that AtVHA-d2 expression was significantly induced by multiple stresses, especially oxidative stress. Phenotypic analysis revealed that the atvha-d2 mutant was sensitive to oxidative stress, and oxidative stress resulted in changes in root H+ flux of the atvha-d2 mutant. RNA-seq combined with qPCR analysis showed that the expression of several members of AtAHA in the atvha-d2 mutant under oxidative stress was different from that in the wild-type, and the differential expression of AtAHA genes might be related to the change in root H+ flux in the atvha-d2 mutant. In conclusion, our results indicate that AtVHA-d2 plays a role in Arabidopsis in response to oxidative stress by affecting H+ flux and AtAHA expression.”
Open Review #2
Comments and Suggestions for Authors
- Comment:It suggest to add information about the reasons of the chosen different concentration of chemicals or range of temperature used to induce stresses. Is there a some links between such chosen ranges of stresses e.g. the similar cell sap osmolarity, etc.? Why authors used different concentration of the same chemicals for presented analyses in the manuscript; e.g. (a) 5 mM H2O2for Quantitative real-time PCR analyses, (b) 1 and 2 mM H2O2 for studies of salt tolerant phenotype, (c) 1, 2 and 3 mM H2O2 for hydrogen peroxide content measurements.
Response: Thank you for your comment. Phenotypic analyses showed that Arabidopsis seeds and seedlings had different tolerance to multiple abiotic stresses. For example, Arabidopsis seeds can hardly germinate on 1/2 MS medium containing 5 mM H2O2. However, Arabidopsis seedlings (7-day-old) transferred to to 1/2 MS containing 5 mM H2O2 can still grow in a short period. (a) Low concentrations of H2O2 may not cause significant changes in AtVHA-d gene expression, thus, we selected high concentrations of H2O2 (5 mM) to treat Arabidopsis seedlings for qPCR analyses. (b) For phenotypic analysis, we selected low concentrations of H2O2 (1, 2, or 3 mM) for treatment. At this concentration, Arabidopsis seeds can germinate normally, but seedling growth can be significantly inhibited. (c) The 1, 2, and 3 mM H2O2 were selected for the measurement of H2O2 content to correspond to concentrations used in phenotypic analysis. Concentration selection of other abiotic stress factors (NaCl and mannitol) was similar to oxidative stress. The 150 mM NaCl and 300 mM mannitol was used for qPCR analysis, while 75 and 100 mM NaCl, 175 and 200 mM mannitol were used for phenotypic analysis.
We have added some explanations to the text of the revised manuscript. (Page 12, lines 306-315).
“Studies have shown that V-ATPase is involved in multiple stress responses [12, 22, 29]. The expression of AtVHA-d genes was investigated under abiotic stress conditions that significantly inhibited the growth of Arabidopsis seedlings, including 150 mM NaCl, 300 mM mannitol, 5 mM H2O2, 100 µM ABA, cold (4 °C), and heat (37 °C). qPCR analysis showed that AtVHA-d2 was significantly induced by multiple stresses, especially oxidative stress (Figure 4). The phenotype of the atvha-d2 mutant and wild-type were compared under the conditions of abiotic stress that moderately inhibited the growth of Arabidopsis seedlings, including 75 and 100 mM NaCl, 175 and 200 mM mannitol, 1 and 2 mM H2O2. Phenotypic analysis showed that the relative root length and fresh weight of atvha-d2 were significantly lower than those of Col-0 under oxidative stress (Figure 6).“
- 2. Comment:On figure e.g. 6, authors showed differences in the growth of whole seedlings (and their roots) Col-0 and atvha-d2 It would be interesting, in the future, to corelate the level of H2O2in different organs such as leaves, steam, roots with the V-ATPaze d2 subunits expression.
Response: Thanks for your suggestion, it is very helpful for our study. In the future, we will continue to investigate the function of AtVHA-d2 in oxidative stress.
Open Review #3
Comments and Suggestions for Authors
- Comment:Supplementary data was not available to me.
Response: We have submitted supplementary data into the submission system. We can also provide supplementary data in other ways, such as e-mail.
Comments:
Background
- 2. Comment:To begin with, the V-ATPase sector V0consists of three to five different subunits depending on the organelle, but of at least seven proteins in total.
Response: Thank you for your comments. We have revised the content to make it more accurate. (Page 2, lines 41-43).
“The V0 domain, which consists of three to five different subunits of six subunits (VHA-a, VHA-c, VHA-c', VHA-c'', VHA-d, VHA-e) on different organelles, is responsible for H+ translocation across membranes. ”.
- 3. Comment:Reversible Dissociation has not been observed in plants, but in yeast and mammals. Schnitzer et al. have even shown that there is no reversible dissociation in Arabidopsis.
Response: Thank you for your comments. We have revised the content and the cited reference. (Page 2, lines 46-47).
“The V1 domain is assembled into a complete enzyme by binding to the V0 domain; this binding is reversible in yeast [5]. The mechanism of this reversible dissociation in plants may be different from that in yeast [7].“
Reference
[7] Schnitzer, D.; Seidel, T.; Sander, T.; Golldack, D.; Dietz, K. J. The cellular energization state affects peripheral stalk stability of plant vacuolar H+-ATPase and impairs vacuolar acidification. Plant Cell Physiol. 2011, 52, 946-956.
- 4. Comment:When dealing with oxidative stress, publications dealing with the redox-regulation of V-ATPase should be included. There are multiple from yeast, Bos taurusand plants. The function of V-ATPase should be seen in the context of oxidizing conditions.
Response: Thank you for your comments. We have added the content on redox-regulation of V-ATPase and cited references. (Page 2, lines 60-64).
“In mammals, fungi, and plants, V-ATPases are subject to oxidative inactivation and activity can be recovered using reducing agents [17-19]. For example, Arabidopsis V-ATPase is reversibly inactivated by hydrogen peroxide (H2O2), and activity can be recovered using reducing agents [20]. However, the regulatory mechanism of V-ATPase in oxidative stress is not clear.”
Reference
[17] Hager, A.; Lanz, C. Essential sulfhydryl groups in the catalytic center of the tonoplast H+-ATPase from coleoptiles of Zea mays L. as demonstrated by the biotin-streptavidin- peroxidase system. Planta 1989, 180, 116–122
[18] Feng, Y.; Forgac, M. Cysteine 254 of the 73-kDa A subunit is responsible for inhibition of the coated vesicle (H+)-ATPase upon modification by sulfhydryl reagents. J. Biol. Chem. 1992, 267, 5817–5822
[19] Dschida, W. J.; Bowman, B. J. The vacuolar ATPase: sulfite stabilization and the mechanism of nitrate inactivation. J. Biol. Chem. 1995, 270, 1557–1563
[20] Seidel, T.; Scholl, S.; Krebs, M.; Rienmuller, F.; Marten, I.; Hedrich, R.; Hanitzsch, M.; Janetzki, P.; Dietz, K. J.; Schumacher, K. Regulation of the V-type ATPase by redox modulation. Biochem. J. 2012, 448, 243-251.
- 5. Comment:The position and function of VHA-d should be considered and discussed. It is amazing that there is no original work cited that deals with VHA-d.
Response: Thank you for your comments. We have added the only results on VHA-d studies and cited references. (Page 2, lines 67-71).
“The Arabidopsis AtVHA-d contains two isoforms, AtVHA-d1 and AtVHA-d2, which are located in close proximity on chromosome three and may have arisen from a gene replication event [21]. Until now, only one study has shown that wheat (Triticum aestivum L.) VHA-d gene expression is induced by salt and abscisic acid [22]. “
Reference
[21] Sze, H.; Schumacher, K.; Muller, M. L.; Padmanaban, S.; Taiz, L., A simple nomenclature for a complex proton pump: VHA genes encode the vacuolar H(+)-ATPase. Trends Plant Sci. 2002, 7, 157-161.
[22] Zhao, Q.; Zhao, Y. J.; Zhao, B. C.; Ge, R. C.; Li, M.; Shen, Y. Z.; Huang, Z. J. Cloning and functional analysis of wheat V-H+-ATPase subunit genes. Plant Mol. Biol. 2009, 69, 33-46.
- 6. Comment:VHA-d and VHA-c aren’t assembly subunits, they are essential subunits of the functional complex.
Response: Thank you for your comments. We have revised this statement. The “important assembly” was revised to be “essential”. (Page 12, line 298).
- 7. Comment:It should be clarified that V-PPase, PM-ATPase and V-ATPase co-regulate the CYTOSOLIC pH.
Response: cWe have revised this statement. The “H+ balance” was revised to be “cytosolic pH”. (Page 12, line 325).
- 8. Comment:VHA-d1 (AT3G28710) and VHA-d2 (AT3G28715) are located in close proximity on chromosome three. This is another hint for gene dublication.
Response: Thank you for your comments. We have revised the content and the cited reference. (Page 2, lines 69-71).
“The Arabidopsis AtVHA-d contains two isoforms, AtVHA-d1 and AtVHA-d2, which are located in close proximity on chromosome three and may have arisen from a gene replication event [21].”
Reference
[21] Sze, H.; Schumacher, K.; Muller, M. L.; Padmanaban, S.; Taiz, L., A simple nomenclature for a complex proton pump: VHA genes encode the vacuolar H(+)-ATPase. Trends Plant Sci. 2002, 7, 157-161.
Methods
- 9. Comment:The materials and methods section is too scarce in general. For instance, microscopy section needs more information such as pixel dwell time, objective, laser line and so on…
Response: Thank you for your comments. We have added this information. (Page 3, lines 103-107).
“Roots of Arabidopsis seedlings stably expressing AtVHA-d2-GFP were washed twice with liquid 1/2 MS medium immediately before visualizing via confocal laser scanning microscopy (Nikon, A1, Tokyo, Japan). Using the 25 × objective lens, the observation area was 1024 × 1024 pixels, and the pixel dwell time was 0.497 μs. GFP signals were detected using a 500–530 nm emission wavelength (FITC) after excitation with 488 nm laser.”
- 10. Comment:One transgenic line was analyzed, that bears an insertion in exon 5 of VHA-d2 (out of ten exons). There is a chance, that a truncated transcript and protein exist. qPCR revealed 10% of WT-transcript amount for atvha-d2, but coverage of the sequences was not analyzed. I further miss a second transgenic line to confirm the observation. Gene editing might even enable to knock out both VHA-d isoforms. However, including a second line would meet the standard.
Response: Thank you for your comments. Our qPCR primers (qF: 5’-CGGAGAATAAGAGG AACCCA-3’; qR: 5’-CACGCAGAGCGACATAAACA-3’;) are located in the 3’-UTR region of the AtVHA-d2 gene. Furthermore, we investigated the expression of AtVHA-d2 gene in multiple atvha-d2 mutant progeny lines, and their expression was similar (WT=1 ± 0.16; atvha-d2: #1: 0.09 ± 0.02; #2: 0.08 ± 0.03; #3: 0.09 ± 0.03; #4: 0.07 ± 0.02). The pre-experimental results show that their phenotypes are similar. Moreover, we used gene editing (CRISPR-Cas9) to knock out AtVHA-d1 gene in the atvha-d2 mutant background. However, we did not obtain effective offspring of gene editing, speculating that simultaneous knockdown of AtVHA-d1 and AtVHA-d12 may be fatal.
- 11. Comment:qPCR is usually done with more than one housekeeping gene. Others should be included besides actin.
Response: Thank you for your comments. In the pre-experiment, we used actin and tubulin as housekeeping genes. Because, the dissolution curve and amplification curve of the qPCR results of actin gene are better, we chose actin as the housekeeping gene.
- 12. Comment:Transgenic lines have been generated, but in which background? Where the GFP-lines generated in the knock down-background?
Response: Thank you for your comments. Background of AtVHA-d2-GFP transgenic lines is wild type. We have added information about “wild-type”. (Page 6, line 186).
- 13. Comment:H2O2was applied for 48 h, I doubt that it lasts longer than a couple of hours. What’s meant by H+-net flux? More information about the method and its informative value is required. A simple reference is not sufficient.
Response: We re-examined the records and confirmed that H2O2 processing time was 48 h. We have added information about H+ flux experiments. (Page 4, lines 137-145).
“Net H+ flux was measured using the Non-invasive Micro-test Technology (NMT100 Series, YoungerUSA LLC, Amherst, MA, USA) as described previously [13, 24, 25]. Seven-day-old seedlings of Col-0 and the atvha-d2 mutant were grown on 1/2 MS medium, and were exposed to mannitol (200 mM), NaCl (100 mM), and H2O2 (2 mM) for 24 h. Root segments were immobilized in the measuring solution (0.1 mM KCl, 0.1 mM CaCl2, 0.1 mM MgCl2, 0.5 mM NaCl, and 0.3 mM MES, pH 5.8) to measure the H+ flux. Each sample was measured continuously for 10 min. The H+ flux rate, based on the voltages monitored between two points (0 and 20 µm), was calculated using iFluxes/imFluxes 1.0 software (Younger USA LLC, Amherst, MA, USA). Six biological repeats were performed for each analysis.”.
- 14. Comment:Subcellular localization: pBI121-GFP does not have a KpnI site and XbaI is located downstream the GFP, resulting in GFP-VHA-d2. Might it be pBI121-GFP2?
Response: We used our previous plasmid [pBI121-(XbaI, BamHI)-PutVHA-c (KpnI)-GFP], which was described in our previous study [14]. We have cited the reference in materials and methods. (Page 3, lines 96-98).
Reference
[15] Zhou, A.; Bu, Y.; Takano, T.; Zhang, X.; Liu, S. Conserved V-ATPase c subunit plays a role in plant growth by influencing V-ATPase-dependent endosomal trafficking. Plant Biotechnol. J. 2016, 14, 271-283.
- 15. Comment:Pattern in the images cannot be interpreted as vacuoles since fully elongated cells are shown. By the way, the V-ATPases is known to be present in all endomembranes of the secretory pathway. Merge of bright field image and GFP-channel does not help much.
Response: We observed GFP signals in cells of the root mature region in which vacuoles are mature central vacuoles. We have revised this description. (Page 12, lines 304-305).
“However, our observations cannot completely exclude AtVHA-d2-GFP localization on other endomembranes.”
Figures
- 16. Comment:Figure 1: What’s the meaning of the color-coding of amino acids?
Response: Thank you for your comments. The same color residues indicate identical residues in each sequence. We have added this content to the legend. (Page 5, lines 173-174).
- 17. Comment:Figure 5b shows an assembled agarose-gel image. That is not acceptable.
Response: The original gel images from Figure 5 b are showed in supplementary Figure S1.
- 18. Comment:Data in Figure 6 is from one experiment. Should be repeated/confirmed independently. Further, it is not clear, if SD or SE is given (same for figure 7).
Response: We are sorry for our negligence. Error bars represent the SE. We have added this content to the legend of Figure 5, 6, 7 and 8. (Pages 7, 8, 9, lines 214-215, 230-231, 234, 250-251).
- 19. Comment:Figure 9: labeling such as “CK” needs to be defined.
Response: Thank you for your comments. CK: control untreated samples. We have added this content to the legend of Figure 9. (Page 10, lines 272).
Editorial comments: Please also add 6-10 more references which were published in recent 3 years.
Response: We have added 8 references and revised the order of references. (Pages 13, 14, lines 458-525).
Other:
1) We re-edited the manuscript for the english language. (please see the full text).

Reviewer 2 Report
It suggest to add information about the reasons of the chosen different concentration of chemicals or range of temperature used to induce stresses. Is there a some links between such chosen ranges of stresses e.g. the similar cell sap osmolarity, etc.? Why authors used different concentration of the same chemicals for presented analyses in the manuscript; e.g. (a) 5 mM H2O2 for Quantitative real-time PCR analyses, (b) 1 and 2 mM H2O2 for studies of salt tolerant phenotype, (c) 1, 2 and 3 mM H2O2 for hydrogen peroxide content measurements.
On figure e.g. 6, authors showed differences in the growth of whole seedlings (and their roots) Col-0 and atvha-d2 mutants. It would be interesting, in the future, to corelate the level of H2O2 in different organs such as leaves, steam, roots with the V-ATPaze d2 subunits expression.
Author Response

(The authors gave the same response as above.)

Reviewer 3 Report
Supplementary data was not available to me.
Comments:
Background
To begin with, the V-ATPase sector Vo consists of three to five different subunits depending on the organelle, but of at least seven proteins in total.
Reversible Dissociation has not been observed in plants, but in yeast and mammals. Schnitzer et al. have even shown that there is no reversible dissociation in Arabidopsis.
When dealing with oxidative stress, publications dealing with the redox-regulation of V-ATPase should be included. There are multiple from yeast, Bos taurus and plants. The function of V-ATPase should be seen in the context of oxidizing conditions.
The position and function of VHA-d should be considered and discussed. It is amazing that there is no original work cited that deals with VHA-d.
VHA-d and VHA-c aren’t assembly subunits, they are essential subunits of the functional complex.
It should be clarified that V-PPase, PM-ATPase and V-ATPase co-regulate the CYTOSOLIC pH.
VHA-d1 (AT3G28710) and VHA-d2 (AT3G28715) are located in close proximity on chromosome three. This is another hint for gene dublication.
Methods
The materials and methods section is too scarce in general. For instance, microscopy section needs more information such as pixel dwell time, objective, laser line and so on…
One transgenic line was analyzed, that bears an insertion in exon 5 of VHA-d2 (out of ten exons). There is a chance, that a truncated transcript and protein exist. qPCR revealed 10% of WT-transcript amount for atvha-d2, but coverage of the sequences was not analyzed. I further miss a second transgenic line to confirm the observation. Gene editing might even enable to knock out both VHA-d isoforms. However, including a second line would meet the standard.
qPCR is usually done with more than one housekeeping gene. Others should be included besides actin.
Transgenic lines have been generated, but in which background? Where the GFP-lines generated in the knock down-background?
H2O2 was applied for 48 h, I doubt that it lasts longer than a couple of hours.
What’s meant by H+-netflux? More information about the method and its informative value is required. A simple reference is not sufficient.
Subcellular localization: pBI121-GFP does not have a KpnI site and XbaI is located downstream the GFP, resulting in GFP-VHA-d2. Might it be pBI121-GFP2?
Pattern in the images cannot be interpreted as vacuoles since fully elongated cells are shown. By the way, the V-ATPases is known to be present in all endomembranes of the secretory pathway. Merge of bright field image and GFP-channel does not help much.
Figures
Figure 1: What’s the meaning of the color-coding of amino acids?
Figure 5b shows an assembled agarose-gel image. That is not acceptable.
Data in Figure 6 is from one experiment. Should be repeated/confirmed independently. Further, it is not clear, if SD or SE is given (same for figure 7)
Figure 9: labeling such as “CK” needs to be defined.
Author Response

(The authors gave the same response as above.)

Round 2
Reviewer 3 Report
Most points of criticism have been addressed somehow, however, some remained:
VHA-c' is not a subunit of the plant V-ATPase
References for VHA-d: I miss Thaker et al. 2007 dealing with VHA-d from yeast, showing the structure and importance of disulfide formation for folding.
Methods and microscopy: Please name the objective, it is the most important part of any microscope, provide the full description including NA, immersion, correction etc...
Measurement of Net H+-flux: The improvement does not help much without digging into cited literature. State at least the flux direction that is measured.
Figure 1: coloring corresponds to properties of amino acids, doesn't it? Acidic ones are red and so on...
Figure 3: Cannot find the improvements you said you did
Methods: Cloning, refer to the precise plasmid you have used. I understand that it was already modified and no longer identical to pBI121-GFP
Showing for instance embryo-lethality of gene editing of vha-d1 in the vha-d2 KD-background would have been a vast improvement...
Author Response
18/06/2020
Genes
Dear Editor:
We wish to resubmit our manuscript entitled “Arabidopsis V-ATPase d2 Subunit Plays a Role in Plant Responses to Oxidative Stress” (manuscript ID: genes-830882). We are indeed thankful for the valuable comments, which were very helpful in improving our manuscript. We have perused the comments carefully and have made the suggested corrections.
We are hopeful that the revised manuscript meets your approval. The revised portions are marked in red. The major corrections done in the manuscript and the point-wise response to the reviewers’ comments are presented at the end of this letter.
We have made an earnest effort to incorporate all the suggested changes in the revised manuscript and look forward to a favorable decision from you.
Sincerely,
[Author’s name] Shuang Feng and Yuanyuan Bu
[Affiliation] Key Laboratory of Saline-Alkali Vegetation Ecology Restoration (Northeast Forestry University), Ministry of Education, Harbin 150040, PR China
[Postal address] Hexing Road No. 26, Xiangfang District, Harbin City, Heilongjiang Province 150040, China.
[Phone number] +86 451 8219 2763
[Fax number] +86 451 8219 2763
[Email address] fengshuang86@163.com; yuanyuanbu@nefu.edu.cn
Response to the reviewers’ comments
Open Review #3
Comments and Suggestions for Authors
- Comment:Most points of criticism have been addressed somehow, however, some remained:
Response: Thank you for your comments, which are helpful for the improvement of our manuscript.
- 2. Comment:VHA-c' is not a subunit of the plant V-ATPase
Response: Thank you for your comment. We have revised this description. (Page 2, line 42).
“six subunits (VHA-a, VHA-c, VHA-c', VHA-c'', VHA-d, VHA-e)” was changed to “ five subunits (VHA-a, VHA-c, VHA-c'', VHA-d, VHA-e)”.
- 3. Comment:References for VHA-d: I miss Thaker et al. 2007 dealing with VHA-d from yeast, showing the structure and importance of disulfide formation for folding.
Response: Thank you for your comment. We have cited this reference. Furthermore, we have cited another study on yeast VHA-d. (Page 2, lines 67-71).
“In yeast, VHA-d was found to be likely involved in the coupling of proton transport and ATP hydrolysis [21]. Furthermore, structural stability of VHA-d depends on the formation of a disulfide bond [22]. In plant, only one study has shown that wheat (Triticum aestivum L.) VHA-d gene expression is induced by salt and abscisic acid [23].”.
[21] Thaker, Y. R.; Roessle, M.; Grüber. G. The boxing glove shape of subunit d of the yeast V-ATPase in solution and the importance of disulfide formation for folding of this protein. J. Bioenerg. Biomembr. 2007, 39, 275-289.
[22] Owegi, M. A.; Pappas, D. L.; Finch, M.W. Jr.; Bilbo, S. A.; Resendiz, C. A.; Jacquemin, L. J.; Warrier, A.; Trombley, J. D.; McCulloch, K. M.; Margalef, K. L.; Mertz, M. J.; Storms, J. M.; Damin, C. A.; Parra, K. J. Identification of a domain in the V0 subunit d that is critical for coupling of the yeast vacuolar proton-translocating ATPase. J. Biol. Chem. 2006, 281, 30001-30014.
- 4. Comment:Methods and microscopy: Please name the objective, it is the most important part of any microscope, provide the full description including NA, immersion, correction etc...
Response: We have added this information. (Page 3, lines 107-108).
“Using the 40 × oil-immersion objective lens (Plan Apochromat; numerical aperture 1.3), the observation area was 1024 × 1024 pixels, and the pixel dwell time was 0.497 μs. ”.
- 5. Comment:Measurement of Net H+-flux: The improvement does not help much without digging into cited literature. State at least the flux direction that is measured.
Response: We have added more detailed information. (Page 4, lines 143-154).
“Root segments were immobilized in the measuring solution (0.1 mM KCl, 0.1 mM CaCl2, 0.1 mM MgCl2, 0.5 mM NaCl, and 0.3 mM MES, pH 5.8) to measure the H+ flux. The roots were fixed to the bottom of the plate using resin blocks and filter paper strips. Each sample was measured continuously for 10 min. To take flux measurements, the ion-selective electrodes were calibrated using pH 5.5, 6.0, and 6.5 solutions, respectively. The H+ flux rate, based on the voltages monitored between two points (0 and 20 µm), was calculated using iFluxes/imFluxes 1.0 software (Younger USA LLC, Amherst, MA, USA). The calibration slope for H+ was 54.42 mV/decade. Six biological repeats were performed for each analysis.
Ion flux was calculated by Fick’s law of diffusion:
J= −D (dc / dx) |
where J represents the ion flux in the x direction, dc/dx is the ion concentration gradient, and D is the ion diffusion constant in a particular medium.”.
- 6. Comment:Figure 1: coloring corresponds to properties of amino acids, doesn't it? Acidic ones are red and so on...
Response: The same color residues indicate identical residues in each sequence. We have explained in the legend. (Page 5, lines 182-183).
- 7. Comment:Figure 3: Cannot find the improvements you said you did
Response: We have revised Figure 3. (Page 6, lines 197-199).
- 8. Comment:Methods: Cloning, refer to the precise plasmid you have used. I understand that it was already modified and no longer identical to pBI121-GFP
Response: Thank you for your comment. We have revised this description. (Page 3, line 100).
“pBI121-GFP” was changed to “pBI121-PutVHA-c-GFP”.
- 9. Comment:Showing for instance embryo-lethality of gene editing of vha-d1 in the vha-d2 KD-background would have been a vast improvement...
Response: Thank you for your comment. We speculate that simultaneous knockdown of AtVHA-d1 and AtVHA-d12 may be fatal. In the future, we plan to use VIGS methods to silence VHA-d1 in the atvha-d2 mutant background. If the material can be obtained, we will continue to study the function of the VHA-d.
Other:
1) We have added 2 references [21 and 22] and revised the order of references. (Page 14, lines 416-454).
